# Identifying Localized Pollution Sources in a Two-Dimensional Diffusion–Reaction Equation via Conjugate Gradient with Iterative Regularization

**Pronin K. A.**[1][*]      **Krivorotko O. I.**[2]

[1,2] Sirius University, Sochi, Russia

## Abstract

The inverse problem of recovering time-dependent intensities of localized pollution sources from pointwise concentration measurements is considered. The forward model is a two-dimensional parabolic diffusion–reaction equation with homogeneous Neumann boundary conditions and a nonzero initial condition representing pre-existing contamination. Point sources are approximated by bilinear distributions on the computational grid, placing the problem within the standard $L^2$-framework. Discrete adjoint equations are derived via the Lagrangian (discretize-then-optimize), yielding an explicit gradient requiring one forward and one adjoint solve. The inverse problem is solved by the Polak–Ribière conjugate gradient method with exact line search; regularization is achieved by early stopping via the Morozov discrepancy principle. Spectral analysis of the discrete observation operator ($\sigma_k \sim k^{-4.7}$, condition number $\sim 10^{15}$, effective rank 16) quantitatively explains the reconstruction errors and the effectiveness of early stopping. Numerical experiments with 1–10% multiplicative noise confirm that the discrepancy principle terminates iterations correctly ($\rho/\delta \in [1.00, 1.07]$), yielding errors of 36–45%. Iterative regularization matches optimally tuned Tikhonov regularization without parameter selection.

## 1 Introduction

### 1.1 Motivation

Identification of pollution sources from indirect concentration measurements is a fundamental problem in environmental monitoring. In practice, a contaminant enters the atmosphere or a water body from several localized sources, while a network of stationary sensors records concentrations at only a finite number of control points. This gives rise to an *inverse problem*: given incomplete and noisy data, recover the source characteristics—locations, strengths, and time-dependent intensity profiles.

The transport of a passive scalar is governed by a parabolic PDE. Determining its right-hand side from indirect data is *ill-posed* in the sense of Hadamard (Kabanikhin, 2011; Isakov, 2006): small data perturbations can cause arbitrarily large errors, necessitating *regularization*. Related settings arise in inverse heat conduction (Pyatkov & Rotko, 2020) and epidemiological modelling (Murray, 2002).

### 1.2 Literature review

**Theoretical foundations.**  The general theory is presented in Kabanikhin (2011), Isakov (2006), and Engl et al. (1996). For parabolic equations with point sources, El Badia & Ha-Duong (2002) proved that $2N$ sensors in general position suffice for unique determination of $N$ sources. For *known* locations (the present setting), $M \geq N$ sensors suffice provided injectivity holds. Well-posedness with $\delta$-sources in anisotropic Sobolev spaces was established in Pyatkov & Safonov (2017).

---

[*]Corresponding author.

**Computational methods.** Adjoint-based optimization for source identification was developed by Penenko (2019) for ODE systems and applied to passive tracer transport in Kochergin & Kochergin (2015). The present work extends this approach to a full 2D PDE model with a self-contained exposition of all discretization steps.

**Regularization.** Classical Tikhonov regularization (Tikhonov & Arsenin, 1977) requires parameter selection (Hansen, 1998). An alternative is *iterative regularization*: CG applied to linear ill-posed problems possesses the semi-convergence property (Nemirovskii, 1986; Hanke, 1995), with iteration number $k^*$ acting as $1/\alpha$, determined by the Morozov discrepancy principle (Morozov, 1966).

### 1.3 CONTRIBUTIONS

In contrast to Penenko (2019) (ODE systems) and Kochergin & Kochergin (2015) (geophysical application without spectral analysis), this work provides a *self-contained computational pipeline*:

1. Forward and inverse problems for a 2D diffusion–reaction equation with bilinear source approximation and Gaussian initial condition.
2. Discrete adjoint equations and explicit gradient via discretize-then-optimize, with exact consistency.
3. Spectral analysis of the discrete observation operator (SVD, Picard condition, energy distribution), giving a quantitative explanation of reconstruction errors.
4. Demonstration that iterative regularization matches optimally tuned Tikhonov without parameter selection.
5. Exact analytical line search (not available in Penenko (2019) or Kochergin & Kochergin (2015)), guaranteeing monotone decrease of $J^h$.

### 1.4 OUTLINE

Sections 2–5 contain the mathematical formulation, discretization, adjoint derivation, and algorithm. Section 6 reports numerical experiments. Section 7 gives conclusions.

## 2 MATHEMATICAL FORMULATION

### 2.1 FORWARD PROBLEM

Let $\Omega = (0, L_x) \times (0, L_y) \subset \mathbb{R}^2$, $T > 0$, $Q_T := \Omega \times (0, T]$. Consider

$$\frac{\partial u}{\partial t} = a^2 \Delta u - \mu\, u + \sum_{j=1}^{N} q_j(t)\, \phi_h(x - x_j, y - y_j), \quad (x, y, t) \in Q_T, \tag{1}$$

$$u(\cdot, 0) = u_0, \quad \left.\frac{\partial u}{\partial \mathbf{n}}\right|_{\partial\Omega} = 0, \quad t \in (0, T], \tag{2}$$

where $a^2 > 0$, $\mu \geq 0$, $(x_j, y_j)$ are known source positions, $q_j \in L^2(0, T)$ are unknown intensities, and $\mathbf{n}$ is the outward normal.

**Source approximation.** Each point source is replaced by a *bilinear kernel* $\phi_h$ satisfying: (R1) $\int_\Omega \phi_h = 1$; (R2) $\operatorname{supp}(\phi_h) \subset [0, h_x] \times [0, h_y]$; (R3) $\phi_h \in L^\infty(\Omega)$; (R4) $\phi_h \to \delta$ distributionally as $h \to 0$. By (R1)–(R3), the right-hand side of equation 1 lies in $L^2(0, T; L^2(\Omega))$, ensuring standard parabolic theory applies (Evans, 2010).

**Initial condition.**

$$u_0(x, y) = \sum_{j=1}^{N} A_j \exp\left(-\frac{(x - x_j)^2 + (y - y_j)^2}{2\sigma_j^2}\right), \quad \sigma_j = \sqrt{2a^2 t_{\text{pre}}}, \tag{3}$$

modelling contamination from sources active before $t = 0$. This coincides with the fundamental solution of the diffusion equation at time $t_{\text{pre}}$ (Evans, 2010). The initial condition is *known* and its effect is eliminated by subtraction (see below).

## 2.2 Observation operator and inverse problem

Sensors at $(\hat{x}_i, \hat{y}_i) \neq (x_j, y_j)$ (non-collocation required (El Badia & Ha-Duong, 2002)). By linearity, $u[q] = u_{\text{hom}} + \widetilde{u}[q]$, where $u_{\text{hom}}$ solves equation 1–equation 2 with $q \equiv 0$. Define

$$\mathcal{S}q := \widetilde{u}[q], \quad \mathcal{C}v := \{v(\hat{x}_i, \hat{y}_i, \cdot)\}_{i=1}^{M}, \quad \mathcal{A} := \mathcal{C} \circ \mathcal{S} : \mathcal{Q} \to \mathcal{F}, \tag{4}$$

with $\mathcal{Q} = (L^2(0,T))^N$, $\mathcal{F} = (L^2(0,T))^M$.

**Proposition 1.** $\mathcal{A}$ *is linear, bounded, compact, and (under the condition of El Badia & Ha-Duong (2002), $M \geq N$) injective.*

*Proof sketch. Linearity/boundedness*: direct from definitions. *Compactness*: by parabolic regularity (Evans, 2010, §7.1), $\mathcal{S}$ maps into $W := L^2(0,T; H^2) \cap H^1(0,T; L^2)$; the Aubin–Lions theorem (Simon, 1987) gives $W \hookrightarrow\hookrightarrow L^2(0,T; H^s)$ for $s < 2$; Morrey's embedding $H^s(\Omega) \hookrightarrow C(\bar{\Omega})$ ($d = 2$, $s > 1$) then makes $\mathcal{C} \circ \mathcal{S}$ compact. *Injectivity*: El Badia & Ha-Duong (2002, Thm. 2.1). $\square$

Compactness implies ill-posedness: $\mathcal{A}^{-1}$ is unbounded. The decay rate of singular values $\sigma_k(\mathcal{A}_h)$ is analysed in Appendix B.

**Data model and correction.** Raw sensor data: $f_i^{\text{raw}} = u[q^*](\hat{x}_i, \hat{y}_i, \cdot) + \varepsilon_i$. After subtracting $u_{\text{hom}}$ (one forward solve):

$$f_i := f_i^{\text{raw}} - u_{\text{hom}}(\hat{x}_i, \hat{y}_i, \cdot) = (\mathcal{A}q^*)_i + \varepsilon_i, \quad \|\varepsilon\|_{\mathcal{F}} = \delta_{\text{noise}}. \tag{5}$$

The inverse problem is to minimise $J[q] := \frac{1}{2}\|\mathcal{A}q - f\|_{\mathcal{F}}^2$, which is quadratic, convex, and (by injectivity) has a unique minimiser.

## 3 Discretization of the forward problem

### 3.1 Grids, Laplacian, and source vectors

Uniform grids: $x_m = mh_x$, $y_n = nh_y$, $t^k = k\tau$; $\mathbf{u}^k \in \mathbb{R}^{N_h}$, $N_h = (N_x+1)(N_y+1)$, lexicographic ordering $I(m,n) = n(N_x+1) + m$. The discrete Laplacian $L = I_{N_y+1} \otimes D_{xx} + D_{yy} \otimes I_{N_x+1}$ is symmetric, negative semi-definite, with $\ker(L) = \text{span}\{\mathbf{1}\}$.

Source vector $\mathbf{d}_j$ has four nonzero bilinear weights divided by $h_x h_y$:

$$(\mathbf{d}_j)_{I(m_j+p, n_j+q)} = \frac{[\xi_j^p(1-\xi_j)^{1-p}][\eta_j^q(1-\eta_j)^{1-q}]}{h_x h_y}, \quad p,q \in \{0,1\}, \tag{6}$$

with $\xi_j = (x_j - m_j h_x)/h_x$, $\eta_j = (y_j - n_j h_y)/h_y$, and $\sum_{m,n}(\mathbf{d}_j)_{m,n} h_x h_y = 1$. Sensor vector $\mathbf{c}_i$ contains bilinear weights *without* $1/(h_x h_y)$; this asymmetry is required for the gradient formula equation 11.

### 3.2 Implicit Euler scheme

$$A_\tau \mathbf{u}^{k+1} = \mathbf{u}^k + \tau \sum_{j=1}^{N} q_j^{k+1} \mathbf{d}_j, \quad A_\tau := (1+\tau\mu)I - \tau a^2 L. \tag{7}$$

**Proposition 2.** $A_\tau$ *is symmetric positive definite ($\mathbf{v}^T A_\tau \mathbf{v} \geq (1+\tau\mu)\|\mathbf{v}\|^2 > 0$ since $L \preceq 0$), so equation 7 is unconditionally stable. A single sparse LU factorisation of $A_\tau$ serves all time steps and the adjoint problem.*

Approximation order: $O(\tau + h_x^2 + h_y^2)$. Implicit Euler is preferred over Crank–Nicolson for (a) symmetric adjoint ($A_\tau = A_\tau^T$), (b) monotonicity for non-smooth sources (Thomée, 2006), and (c) sufficient accuracy at $\tau = 0.05$.

## 4 ADJOINT PROBLEM AND GRADIENT

The discrete functional is

$$J^h[\mathbf{q}] = \frac{\tau}{2} \sum_{i=1}^{M} \sum_{k=1}^{K} (\mathbf{c}_i^T \mathbf{u}^k - f_i^k)^2. \tag{8}$$

The Lagrangian with forward equations as constraints:

$$\mathcal{L} = J^h + \sum_{k=0}^{K-1} (\boldsymbol{\psi}^{k+1})^T (A_\tau \mathbf{u}^{k+1} - \mathbf{u}^k - \tau \sum_j q_j^{k+1} \mathbf{d}_j). \tag{9}$$

Stationarity $\partial \mathcal{L} / \partial \mathbf{u}^l = 0$ gives the adjoint recurrence (backward in time):

$$A_\tau \boldsymbol{\psi}^l = \boldsymbol{\psi}^{l+1} - \mathbf{r}^l, \quad l = K, \dots, 1 \quad (\boldsymbol{\psi}^{K+1} := \mathbf{0}), \tag{10}$$

where $\mathbf{r}^l := \tau \sum_{i=1}^{M} (\mathbf{c}_i^T \mathbf{u}^l - f_i^l) \mathbf{c}_i$. Since $A_\tau = A_\tau^T$, the *same* LU factorisation is reused. Differentiating $\mathcal{L}$ w.r.t. $q_j^l$:

$$g_j^l := \frac{\partial J^h}{\partial q_j^l} = -\tau \mathbf{d}_j^T \boldsymbol{\psi}^l, \quad j = 1, \dots, N, \quad l = 1, \dots, K. \tag{11}$$

Cost: $2K$ linear solves (forward + adjoint). Pseudocode in Appendix A.

### 4.1 VERIFICATION

**Mass conservation ($\mu = 0$).**   Relative error $\leq 2.3 \times 10^{-15}$ over $K = 200$ steps: machine precision, confirming conservativeness of the bilinear approximation.

**Superposition.**   $\max_{i,k} |f_{\text{corrected},i}^k - f_{\text{forced},i}^k| = 7.2 \times 10^{-16}$.

**Taylor test.**   Since $J^h$ is quadratic, the expansion

$$J^h(\mathbf{q} + \varepsilon \delta \mathbf{q}) = J^h(\mathbf{q}) + \varepsilon \langle \nabla J^h, \delta \mathbf{q} \rangle + \frac{\varepsilon^2}{2} \|\mathcal{A}_h \delta \mathbf{q}\|_\tau^2 \tag{12}$$

is *exact*, so $\eta_1(\varepsilon)/\eta_1(\varepsilon/2) = 4$ exactly. Confirmed numerically ($4.00 \pm 10^{-12}$, 15 decades).

## 5 INVERSE PROBLEM ALGORITHM

The functional equation 8 is minimised by CG with the Polak–Ribière update. Three properties motivate this choice: (1) *semi-convergence* (Hanke, 1995; Nemirovskii, 1986): early iterations recover components with $\sigma_j \gg \delta_{\text{noise}}$; later iterations amplify noise; (2) *PR automatic restarts* ensuring descent (Nocedal & Wright, 2006); (3) *exact line search*: since $\mathcal{A}_h$ is linear, $\varphi(\gamma) = J^h[\mathbf{q}^{(k)} + \gamma \mathbf{s}^{(k)}]$ is quadratic, minimised analytically at

$$\gamma^{(k)} = -\varphi'(0)/\varphi''(0), \quad \varphi'(0) = \tau \sum_{i,l} r_i^l w_i^l, \quad \varphi''(0) = \tau \sum_{i,l} (w_i^l)^2, \tag{13}$$

where $r_i^l = \mathbf{c}_i^T \mathbf{u}^l - f_i^l$, $w_i^l = \mathbf{c}_i^T \mathbf{w}^l$, and $\{\mathbf{w}^l\}$ solves the forward problem with sources $\mathbf{s}^{(k)}$.

**Iteration:** update $\mathbf{q}^{(k+1)} = \mathbf{q}^{(k)} + \gamma^{(k)} \mathbf{s}^{(k)}$; compute gradient equation 11; set $\beta^{(k+1)} = \max(\langle \mathbf{g}^{(k+1)}, \mathbf{g}^{(k+1)} - \mathbf{g}^{(k)} \rangle / \|\mathbf{g}^{(k)}\|^2, 0)$; update $\mathbf{s}^{(k+1)} = -\mathbf{g}^{(k+1)} + \beta^{(k+1)} \mathbf{s}^{(k)}$. Cost per iteration: $2K$ solves (not available in Penenko (2019) or Kochergin & Kochergin (2015)).

**Morozov stopping criterion.**   Terminate when

$$\rho^{(k)} := \|\mathcal{A}_h \mathbf{q}^{(k)} - \mathbf{f}\|_\tau \leq \eta \, \delta_{\text{noise}}, \quad \eta = 1.1, \tag{14}$$

where $\|\mathbf{v}\|_\tau := (\tau \sum_{i,l} v_{i,l}^2)^{1/2}$ (Engl et al., 1996). When $\delta_{\text{noise}} = 0$, use gradient criterion $\|\mathbf{g}^{(k)}\| \leq \varepsilon_g \|\mathbf{g}^{(0)}\|$.

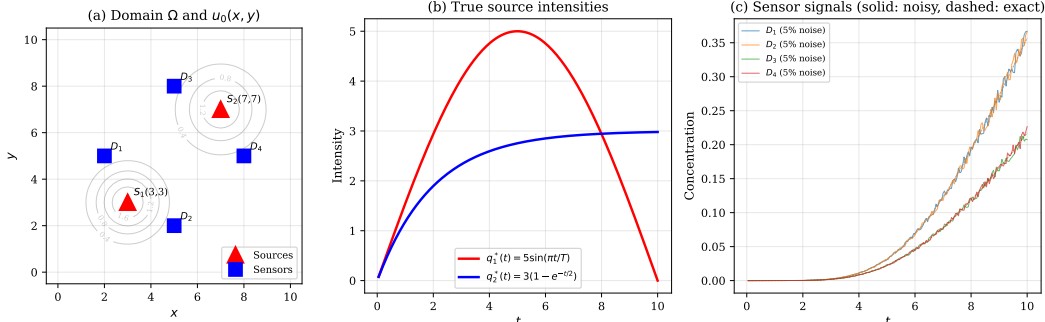

Figure 1: (a) Domain $\Omega$: initial condition contours, sources (▲), sensors (■). (b) True source intensities. (c) Sensor signals: solid—5% noise, dashed—exact.

Table 1: Results for noisy data ($\alpha = 0$, discrepancy principle).

| $\delta_n$ | $\|\varepsilon\|_\tau$ | $k^*$ | $\rho$ | $\rho/\delta$ | Error |
|---|---|---|---|---|---|
| 0 | — | 31 | $2.52 \times 10^{-5}$ | — | 26.5% |
| 1% | $4.28 \times 10^{-3}$ | 7 | $4.33 \times 10^{-3}$ | 1.01 | 35.6% |
| 5% | $2.14 \times 10^{-2}$ | 4 | $2.28 \times 10^{-2}$ | 1.07 | 44.5% |
| 10% | $4.28 \times 10^{-2}$ | 4 | $4.30 \times 10^{-2}$ | 1.00 | 44.5% |

## 6 NUMERICAL EXPERIMENTS

Experiments address: (1) lower error bound and its origin; (2) effectiveness of the Morozov criterion; (3) comparison with Tikhonov regularization. All experiments use synthetic data with known $\mathbf{q}^{\text{true}}$.

### 6.1 MODEL PARAMETERS

Parameters: $\Omega = (0, 10)^2$, $a^2 = 0.1$ (Okubo, 1971), $\mu = 0.01$ (Schnoor, 1996), $T = 10$, grid $51 \times 51$, $K = 200$, $\tau = 0.05$ (Fo = 1.25); source 1 at $(3, 3)$: $q_1^*(t) = 5 \sin(\pi t/T)$; source 2 at $(7, 7)$: $q_2^*(t) = 3(1 - e^{-t/2})$; initial condition equation 3 with $A_1 = 2$, $\sigma_1 = 1.0$, $A_2 = 1.5$, $\sigma_2 = 1.2$; sensors ($M = 4$): $(2, 5), (5, 2), (5, 8), (8, 5)$; noise: $f_i^\delta = f_i(1 + \delta_n \xi_i)$, $\xi_i \sim U[-1, 1]$ (Penenko, 2019). The domain is illustrated in Fig. 1.

### 6.2 BASELINE: EXACT DATA ($\delta = 0$)

In 31 iterations, CG reaches error 26.5%, recovering 94.9% of the energy of $\mathbf{q}^{\text{true}}$ (Appendix B). The remaining 5.1% lies in components with $\sigma_k < 10^{-7}$, unattainable even with exact data. This reflects the fundamental limitation: the data carry information about only 16 singular components (effective rank at threshold $10^{-3}\sigma_1$) out of $\mathbf{q} \in \mathbb{R}^{400}$.

### 6.3 SOURCE RECONSTRUCTION FROM NOISY DATA

Relative error: $\text{err}(\mathbf{q}^{(k)}) := \|\mathbf{q}^{(k)} - \mathbf{q}^{\text{true}}\|_\tau / \|\mathbf{q}^{\text{true}}\|_\tau \times 100\%$. Results are given in Table 1; reconstructed sources in Fig. 2.

**Remark 1.** *Multiplicative noise with $\xi \sim U[-1, 1]$ gives effective level $\|\varepsilon\|_\tau/\|f\|_\tau \approx \delta_n/\sqrt{3}$ since* $\text{Var}[\xi] = 1/3$.

The Morozov criterion terminates correctly ($\rho/\delta \in [1.00, 1.07]$). The coincidence of errors at 5% and 10% is a *mathematically necessary* consequence of the spectral gap: $\sigma_5 = 6.29 \times 10^{-3} < \delta^{(5\%)} = 2.14 \times 10^{-2} < \delta^{(10\%)} = 4.28 \times 10^{-2}$, giving $k_\delta = 2$ for both levels and identical $k^* = 4$ (plateau phenomenon (Hanke, 1995)).

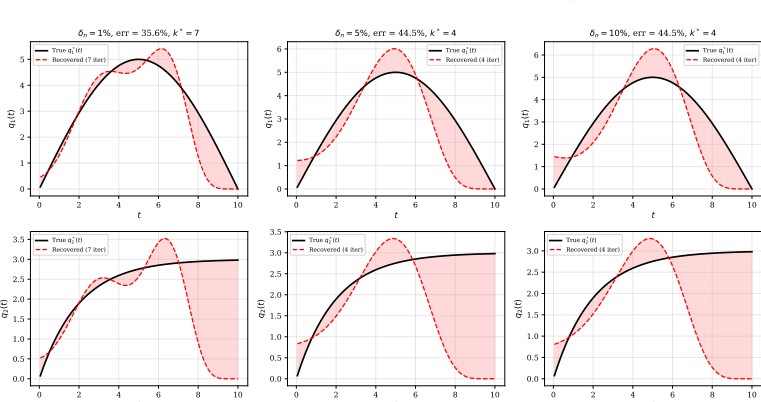

Figure 2: Source reconstruction ($\alpha = 0$, Morozov stopping). Top: $q_1(t)$; bottom: $q_2(t)$. Left to right: $\delta_n = 1\%$ ($k^* = 7$, 35.6%), 5% ($k^* = 4$, 44.5%), 10% ($k^* = 4$, 44.5%). Shading indicates deviation from the true profile.

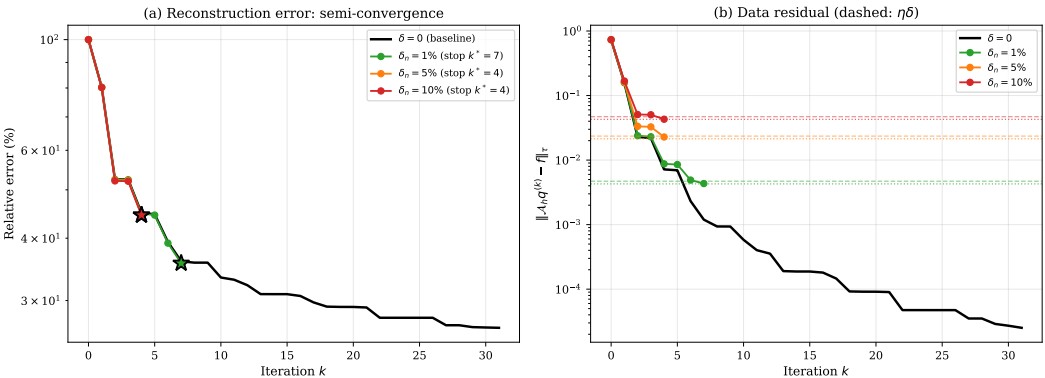

Figure 3: (a) Reconstruction error: noisy curves coincide during early iterations; $\delta = 0$ curve is the lower bound; stars mark Morozov stopping points. (b) Residual curves reaching levels $\eta\delta$ (dashed). (c) Semi-convergence at $\delta_n = 5\%$: error minimum 38.4% at $k = 7$ ($\Diamond$), then divergence.

The systematic reconstruction defects (phase shift of $q_1$ peak by $\Delta t \approx 1.5$–2; boundary underestimation; compensatory overestimation near $t = T/2$) are explained via the spectral structure of $\mathcal{A}_h$ in Appendix B.

## 6.4 CONVERGENCE AND SEMI-CONVERGENCE

Figure 3 shows error and residual dynamics. When CG runs *without stopping* at $\delta_n = 5\%$, the error reaches a minimum of 38.4% at $k = 7$ (oracle optimum) then grows to $\sim$5000% at $k = 100$: for $k > 7$, noise components with $\sigma_k \ll \delta$ are amplified by $\sim 1/\sigma_k$ (at $k = 50$: factor $\sim 2.5 \times 10^5$). The discrepancy principle stops at $k^* = 4$—only 6.1 percentage points above the oracle optimum—without knowledge of $\mathbf{q}^{\mathrm{true}}$.

## 6.5 COMPARISON WITH TIKHONOV REGULARIZATION

The Tikhonov functional $J_\alpha[\mathbf{q}] := \frac{1}{2}\|\mathcal{A}_h \mathbf{q} - \mathbf{f}\|_\tau^2 + \frac{\alpha}{2}\|\mathbf{q}\|_\tau^2$ has minimiser with $k$-th SVD component $\hat{q}_{\alpha,k} = \frac{\sigma_k}{\sigma_k^2 + \alpha}\langle \mathbf{f}, \mathbf{u}_k \rangle_\tau$, where $\mathbf{u}_k$ are the left singular vectors of $\mathcal{A}_h = U\Sigma V^T$. Results for $\delta_n = 5\%$ are in Table 2 and Fig. 4.

Table 2: Effect of Tikhonov parameter $\alpha$ at $\delta_n = 5\%$.

| $\alpha$ | Iter. | Error | $\rho/\delta$ |
|---|---|---|---|
| 0 | 4 | 44.5% | 1.1 |
| $10^{-7}$ | 4 | 44.5% | 1.1 |
| $10^{-6}$ | 4 | 44.5% | 1.1 |
| $10^{-5}$ | 6 | 41.6% | 1.1 |
| $10^{-4}$ | 6 | 51.2% | 2.2 |
| $10^{-3}$ | 5 | 70.3% | 6.9 |
| $10^{-2}$ | 4 | 86.4% | 19.7 |
| $10^{-1}$ | 3 | 97.4% | 31.8 |

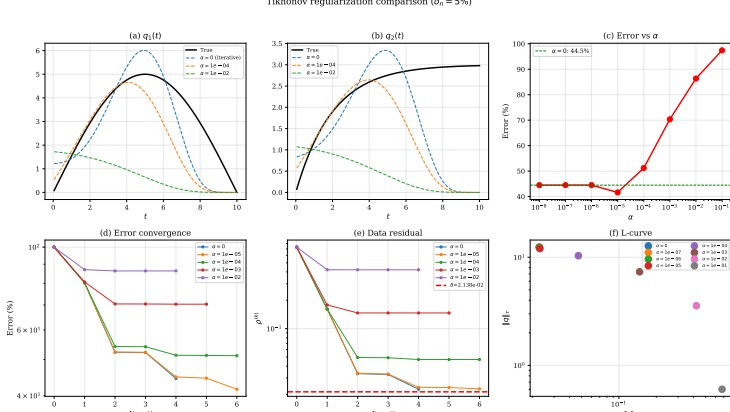

Figure 4: Regularization comparison ($\delta_n = 5\%$). (a)–(b) Reconstructed $q_1$, $q_2$ for $\alpha = 0, 10^{-4}, 10^{-2}$. (c) Error vs. $\alpha$; dashed: $\alpha = 0$ level. (d)–(e) Error and residual vs. iteration. (f) L-curve: corner at $\alpha \approx 10^{-4}$ gives 51.2% error—worse than iterative regularization (44.5%).

Three regimes: *Regime A* ($\alpha \leq 10^{-6}$): penalty negligible, result identical to iterative regularization. *Regime B* ($\alpha = 10^{-5}$): penalty slows convergence, $k^* = 6$, error 41.6% (improvement of 2.9 pp), but $\alpha^*$ found *post factum*. *Regime C* ($\alpha \geq 10^{-4}$): over-regularization; at $\alpha = 10^{-1}$, $\mathbf{q}^{\mathrm{rec}} \approx \mathbf{0}$. The optimal window spans $\sim$0.5 decades; a one-order deviation ($\alpha = 10^{-4}$) raises error to 51.2%.

Iterative regularization (44.5%) matches optimal Tikhonov (41.6%) with a gap of less than 3 pp, without any parameter selection.

## 6.6 CONCENTRATION FIELDS

Figure 5 compares concentration fields at $t = 2.5, 5.0, 10.0$ for $\delta_n = 1\%$.

Despite 35.6% error in intensities, fields at $t \leq 5$ are visually indistinguishable: diffusion suppresses high-frequency components of $q(t)$, and poorly recovered components ($k > 6$) contribute least to the observable field. The large error at $t = T$ is due to boundary underestimation of $q_2$.

## 7 CONCLUSION

1. A discrete 2D diffusion–reaction model with bilinear source approximation and Gaussian initial condition was constructed and verified: unconditional stability proved; mass conservation to $\sim 10^{-15}$.
2. Discrete adjoint equations and gradient equation 11 derived via discretize-then-optimize; correctness confirmed by Taylor test ($4.00 \pm 10^{-12}$, 15 decades).
3. CG–PR with Morozov stopping matches optimally tuned Tikhonov (41.6% vs. 44.5% at $\delta_n = 5\%$) without parameter selection.

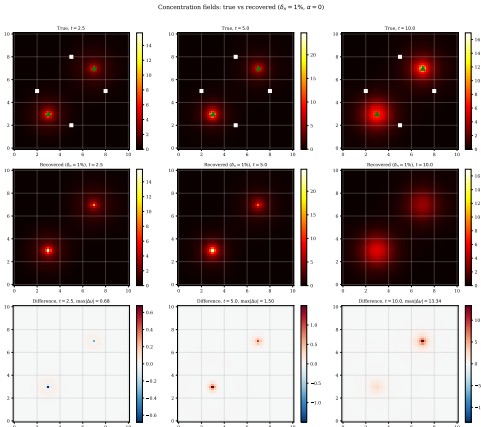

Figure 5: Concentration fields: true (top), reconstructed (middle), difference (bottom); $\delta_n = 1\%$, $\alpha = 0$. $t = 2.5$: $\max|\Delta u| = 0.68$ (4.5%). $t = 5.0$: $\max|\Delta u| = 1.50$ (6.8%). $t = 10.0$: $\max|\Delta u| = 13.3$ (78%, near sources only).

4. Spectral analysis ($\sigma_k \sim k^{-4.7}$, $\kappa \approx 10^{15}$, effective rank 16) quantitatively explained all observed phenomena: the 26.5% lower bound, the error plateau at 5%–10% noise, phase shift, boundary effects, and violation of the Picard condition.

**Limitations and future work.** Source positions are assumed known; joint identification is nonlinear and addressed via gradient extension $\partial\mathcal{L}/\partial x_j = -\tau \sum_l q_j^l (\psi^l)^T \partial\mathbf{d}_j/\partial x_j$. Temporal resolution is limited: $k_\delta \sim (C/\delta)^{1/p}$ with $p \approx 4.7$; halving the error requires $\sim26\times$ noise reduction. Future directions: $H^1$-regularization; optimal sensor placement; extension to convection–diffusion by replacing $A_\tau$ with $A_\tau^{\mathrm{adv}} = (1 + \tau\mu)I - \tau a^2 L + \tau B$ (adjoint then uses $(A_\tau^{\mathrm{adv}})^T$ (Hinze et al., 2009)).

## ACKNOWLEDGMENTS

This work was supported by the grant of the state program of the "Sirius" Federal Territory "Scientific and technological development of the 'Sirius' Federal Territory" (Agreement No. 26-03, date 07.07.2025).

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

## A  GRADIENT COMPUTATION PSEUDOCODE

---

**Input:** $\mathbf{q} = \{q_j^k\}$, $\mathbf{f} = \{f_i^k\}$

1  **Forward** $(k = 0, \ldots, K-1)$: set $\mathbf{u}^0 = \mathbf{0}$; solve $A_\tau \mathbf{u}^{k+1} = \mathbf{u}^k + \tau \sum_j q_j^{k+1} \mathbf{d}_j$; store $\{\mathbf{u}^k\}$.

2  **Residuals** $(k = 1, \ldots, K)$: $\mathbf{r}^k = \tau \sum_i (\mathbf{c}_i^T \mathbf{u}^k - f_i^k) \mathbf{c}_i$.

3  **Adjoint** $(l = K, \ldots, 1)$: solve $A_\tau \boldsymbol{\psi}^l = \boldsymbol{\psi}^{l+1} - \mathbf{r}^l$, $\boldsymbol{\psi}^{K+1} = \mathbf{0}$.

4  **Gradient**: $g_j^l = -\tau \mathbf{d}_j^T \boldsymbol{\psi}^l$.

**Output:** $\mathbf{g} = \nabla_q J^h$.  **Cost:** $2K$ solves with precomputed LU of $A_\tau$.

---

## B  SPECTRAL ANALYSIS OF THE OBSERVATION OPERATOR

The matrix $\mathcal{A}_h \in \mathbb{R}^{800 \times 400}$ is assembled and its SVD computed: $\mathcal{A}_h = U \operatorname{diag}(\sigma_1, \ldots, \sigma_{400}) V^T$. Results: Fig. 6 and Table 3.

**Decay rate.**  $\sigma_k \approx C k^{-p}$: $p \approx 4.7$ ($R^2 = 0.95$, $k = 2 \ldots 50$) and $p \approx 7.7$ ($R^2 = 0.98$, $k = 10 \ldots 100$). Accelerating decay indicates super-algebraic decay characteristic of infinitely smoothing parabolic operators (Engl et al., 1996). Condition number $\kappa \approx 1.2 \times 10^{15}$; effective rank $16/400$ at threshold $10^{-3}\sigma_1$.

Table 3: Singular values of $\mathcal{A}_h$ and Fourier coefficients $\alpha_k := \langle \mathbf{q}^{\text{true}}, \mathbf{v}_k \rangle$.

| $k$ | $\sigma_k$ | $|\alpha_k|$ | $|\alpha_k|/\sigma_k$ | Regime |
|---|---|---|---|---|
| 1 | $9.14 \times 10^{-2}$ | $3.39 \times 10^{1}$ | $3.7 \times 10^{2}$ | signal ($\forall \delta_n$) |
| 5 | $6.29 \times 10^{-3}$ | $1.60 \times 10^{1}$ | $2.5 \times 10^{3}$ | signal ($\delta_n \leq 1\%$) |
| 10 | $1.14 \times 10^{-3}$ | $1.02 \times 10^{1}$ | $9.0 \times 10^{3}$ | noise ($\forall \delta_n$) |
| 20 | $5.02 \times 10^{-5}$ | $8.23 \times 10^{-1}$ | $1.6 \times 10^{4}$ | noise |
| 50 | $8.43 \times 10^{-8}$ | $2.11$ | $2.5 \times 10^{7}$ | noise |
| 100 | $6.92 \times 10^{-11}$ | $1.66 \times 10^{-1}$ | $2.4 \times 10^{9}$ | noise |

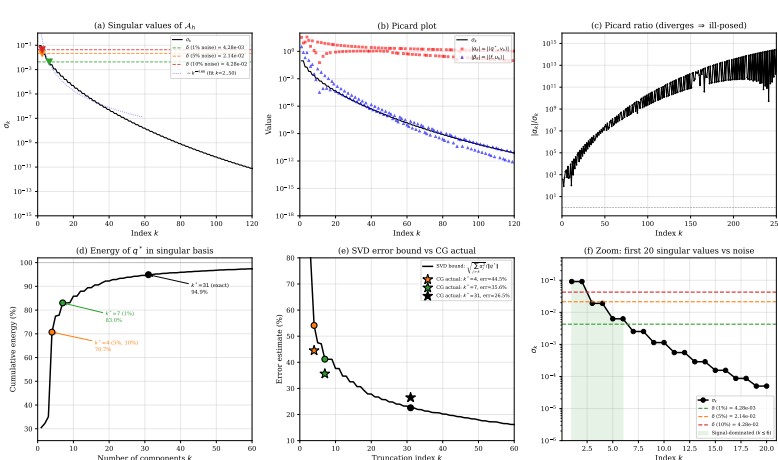

Figure 6: Spectral analysis of $\mathcal{A}_h$. (a) Singular values; dashed: noise levels; dotted: $k^{-4.7}$ fit. (b) Picard plot: $\sigma_k, |\alpha_k|, |\beta_k|$. (c) Ratio $|\alpha_k|/\sigma_k$—grows unboundedly. (d) Cumulative energy; $k^* = 4, 7, 31$ marked. (e) TSVD bound equation 15 and actual CG errors. (f) Zoom: first 20 singular values with noise levels.

**Picard condition.** Coefficients $\alpha_k$ oscillate in $[O(10^{-1}), O(10^{1})]$ and do *not* decay, so $|\alpha_k|/\sigma_k$ grows unboundedly, reaching $\sim 10^{13}$ at $k = 200$. Formal inversion yields amplitudes up to $10^{13}$, confirming the necessity of regularization.

**Truncated-SVD lower bound.**

$$\text{err}_{\text{TSVD}}(k^*) := \sqrt{\frac{\sum_{j>k^*} \alpha_j^2}{\|\mathbf{q}^{\text{true}}\|^2}}. \tag{15}$$

For $k^* = 4$: TSVD bound $54.1\%$, CG error $44.5\%$; $k^* = 7$: $41.2\%$ vs. $35.6\%$; $k^* = 31$: $22.6\%$ vs. $26.5\%$. CG falls *below* the TSVD bound because it minimises $J^h$ over the Krylov subspace $\mathcal{K}_{k^*}(\mathcal{A}_h^T \mathcal{A}_h, \mathcal{A}_h^T \mathbf{f})$; superlinear convergence (Ritz effect (Hanke, 1995)) exploits more than $k^*$ spectral components.

**Resolution index.** $k_\delta := \#\{k : \sigma_k > \delta_{\text{noise}}\}$: $k_\delta = 6$ at $\delta_n = 1\%$; $k_\delta = 2$ at $5\%$ and $10\%$. The discrepancy principle terminates CG at $k^* \approx k_\delta + 2$–$5$ via superlinear convergence, recovering additional components while preventing noise amplification.

