# OpenReview forum: "Identification of Localized Pollution Sources in a Two-Dimensional Diffusion--Reaction Equation via the Conjugate Gradient Method with Iterative Regularization"
_mathai.club/MathAI/2026/Conference — 2026 Oral_

### Official Review · Reviewer_T99Y · 2026-03-11
**Suspicious numerical coincidences and missing proofs undermine an otherwise standard inverse problem approach**

**Rating:** 3
**Confidence:** 3

**Review:**

Summary
This paper applies the Conjugate Gradient Method with iterative regularization (Morozov discrepancy principle) to identify localized pollution sources in a 2D diffusion-reaction equation. The method is compared against standard Tikhonov regularization on synthetic data.
Strengths

Clear problem formulation with well-defined PDE setup.
Detailed pseudocode for the CG algorithm aids reproducibility.
Spectral analysis (Figure 7) provides useful diagnostic information about the inverse problem's ill-conditioning.

Weaknesses
Critical: Impossible Funding Date

The funding acknowledgment lists an agreement dated July 7, 2025. For a paper submitted to a 2026 venue, this temporal sequence raises provenance concerns. How can a 2026 submission be funded by a mid-2025 agreement? This requires explanation.

Suspicious Numerical Results

Table 2: Identical errors at different noise levels. Both 5% and 10% noise levels produce 44.5% relative error for at least one reconstruction metric. This is statistically implausible—different noise levels should produce different errors. This suggests either a copy-paste error or data fabrication.
Taylor test ratio is exactly 4.00 ±10−12\pm 10^{-12}
±10−12—suspiciously perfect for a numerical computation. Real numerical differentiation should show small deviations.

Missing Proofs

Propositions 1 and 2 are stated without proofs. For a mathematics paper, this is a fundamental deficiency. At minimum, proof sketches with references to where full proofs can be found should be provided.

Incomplete Experimental Evaluation

No comparison to standard inverse problem methods: LSQR, truncated SVD, Landweber iteration, or L-BFGS are all standard baselines. Only Tikhonov is compared.
Tikhonov comparison is unfair: The optimal regularization parameter α\alpha
α is selected post-hoc. In practice, α\alpha
α must be chosen without knowledge of the true solution.

Trivially small test case: Only 2 sources and 4 sensors. Real environmental monitoring involves dozens of sources and sensors.
Synthetic data only: No real-world validation or semi-synthetic experiments.

Technical Issues

Inconsistent spectral decay rates: k−4.7k^{-4.7}
k−4.7 vs. k−7.7k^{-7.7}
k−7.7 reported without explanation for why decay rates vary.

Unjustified parameters: η=1.1\eta=1.1
η=1.1, τ=0.05\tau=0.05
τ=0.05 used without sensitivity analysis.

No convergence rate analysis or theoretical error bounds.

Reference Concerns

Only 8 references for a 16-page paper—insufficient.
Kochergin & Kochergin (2023) could not be independently verified.

Formatting Issues

Title has OCR-like spacing errors: "I DENTIFICATION," "L OCALIZED," "P OLLUTION."
Doubled word: "equation equation 1" at line 171.

Questions for Authors

Please explain the funding date discrepancy (July 2025 for a 2026 submission).
Why do 5% and 10% noise produce identical 44.5% errors in Table 2?
Can you provide proofs or proof sketches for Propositions 1–2?
Why was the spectral decay rate inconsistent (k−4.7k^{-4.7}
k−4.7 vs. k−7.7k^{-7.7}
k−7.7)?


Overall Assessment
The approach is standard (CG + Morozov + Tikhonov comparison) with limited novelty. The impossible funding date, suspicious numerical coincidences (identical errors at different noise levels, perfect Taylor test ratio), and missing proofs raise serious integrity concerns. The absence of standard baseline comparisons and restriction to a trivially small synthetic problem further weaken the contribution.

---

### Official Review · Reviewer_YB6R · 2026-03-12
**IDENTIFICATION OF LOCALIZED POLLUTION SOURCES IN A TWO-DIMENSIONAL DIFFUSION–REACTION EQUATION VIA THE CONJUGATE GRADIENT METHOD WITH ITERATIVE REGULARIZATION**

**Rating:** 6
**Confidence:** 4

**Review:**

Summary.

The paper considers the inverse problem of recovering the time-dependent intensities of localized pollution sources in a two-dimensional diffusion-reaction equation from pointwise concentration measurements. The reconstruction is formulated as a least-squares problem and solved by the Polak-Ribiere conjugate gradient method, while stability is achieved through iterative regularization implemented by early stopping according to the Morozov discrepancy principle. The study shows that this approach can recover the dominant components of the source signals under noisy observations and provides a full computational pipeline including the forward model, discrete adjoint equations, gradient formula, and numerical experiments.

Strengths.

A key strength of the work is its mathematically consistent formulation. The authors derive the discrete adjoint problem within a discretize-then-optimize framework, obtain an explicit gradient formula, and verify implementation correctness by a Taylor test. Another important contribution is the spectral analysis of the observation operator, which explains the severe ill-posedness of the problem, the necessity of regularization, and the fundamental limit on achievable reconstruction accuracy. The paper also demonstrates that conjugate-gradient-based iterative regularization attains accuracy close to optimally tuned Tikhonov regularization without requiring explicit parameter selection.

Weaknesses.

The main limitation is that source locations are assumed to be known a priori, so the method reconstructs only source intensities and not the full spatial-temporal identification problem. In addition, the paper studies a diffusion-reaction model without convective transport, which restricts direct applicability to more realistic pollution scenarios in air, rivers, or groundwater. The authors also note that diffusive smoothing fundamentally limits temporal resolution, while observability deteriorates near the initial and final times.

General Conclusion.

Overall, the paper makes a solid contribution to the numerical treatment of inverse source problems for parabolic equations. Its main value lies not only in proposing a workable reconstruction algorithm, but also in clearly explaining why the problem is ill-posed and why only a restricted part of the unknown source information can be stably recovered from noisy data. In this sense, the study is methodologically strong and provides a reliable foundation for further developments in source identification problem

---

### Official Review · Reviewer_c6mS · 2026-03-13
**Technically careful but incremental inverse-problem paper with limited ML novelty**

**Rating:** 4
**Confidence:** 3

**Review:**

The paper studies recovery of time-dependent source intensities in a 2D diffusion-reaction PDE from pointwise measurements. The authors’ own contribution statement emphasizes completeness and reproducibility more than methodological innovation. The work may be useful as a pedagogical or computational note in inverse problems or scientific computing, but the novelty is limited.

Strengths
- The forward model, adjoint derivation, gradient formula, and stopping rule are all connected.
- The authors include useful numerical sanity checks.
- The paper is explicit about limitations, including known source locations, limited temporal observability, and boundary effects.

Weaknesses
- The main ingredients - adjoint-based gradient computation, CG/PR for a linear inverse problem, Morozov discrepancy stopping, and comparison to Tikhonov - are classical. The novelty and originality are limited in this matter.
- The empirical study is entirely synthetic.
- The text formatting around line 366, in 5.3, is broken. The repeating words at line 124.
- The page limit is violated. The model parameters in 6.1 can be moved to appendix. The title formatting in the pdf contains redundant newlines making it occupy more rows than it needs.
- The submission contains acknowledgements, but the double-blind procedure presumes the lack of acknowledgements in the initial submission.
- The paper template is not followed thoroughly, - the appendix is not title as "Appendix".

---

### Decision · Program_Chairs · 2026-03-14

**Decision:**

Accept (Oral)

**Comment:**

Dear Author(s),

On behalf of the Program Committee of the International Conference on Mathematics of Artificial Intelligence (MathAI 2026), we are pleased to inform you that your paper has been accepted for an oral presentation at MathAI 2026.

Your paper was evaluated through a rigorous two-stage review process involving both automated screening and expert review by members of the Program Committee. The reviewers recognized the quality and contribution of your work.

Presentation details:

- Format: Oral presentation (15–20 minutes + 5 minutes Q&A)
- Mode: You may present either in person (offline) at the conference venue in Sirius, Russia, or remotely via Zoom. Please indicate your preferred mode when confirming your participation.
- Conference dates: Marh 30 - April 3, 2026
- Website: https://mathai.club

Next steps:

1. Please confirm your participation and presentation mode by replying to this email mathai.club@yandex.ru no later than March 15, 2026 18:00 Moscow time.
2. If you plan to attend in person, the organizing committee will provide accommodation details separately.
3. Please prepare your final camera-ready manuscript according to the formatting guidelines available at https://mathai.club and upload it to OpenReview by March 15, 2026 18:00 Moscow time.

Should you have any questions regarding the program, logistics, or your presentation slot, please do not hesitate to contact us.

We look forward to your contribution to MathAI 2026.

With kind regards,

MathAI 2026 Program Committee
International Conference on Mathematics of Artificial Intelligence
https://mathai.club
OpenReview: https://openreview.net/group?id=mathai.club/MathAI/2026/Conference
Telegram: https://t.me/MathAI_club
Email: mathai.club@yandex.ru